# Immunogenicity and safety of mRNA-based seasonal influenza vaccines encoding hemagglutinin and neuraminidase

Amanda K. Rudman Spergel[1], Ivan T. Lee[1], Kindra Koslovsky[1], Kristi Schaefers[1], Andrei Avanesov[1], Douglas K. Logan[2], John Hemmersmeier[3], David Ensz[4], Daniel Stadlbauer[1], Bo Hu[1], Alicia Pucci[1], Jignesh Vakil[1], Robert Paris [1], Jintanat Ananworanich[1] & Raffael Nachbagauer [1] ✉

Current influenza vaccines induce immune responses to hemagglutinin (HA), a surface glycoprotein of seasonal influenza viruses, but have suboptimal effectiveness. mRNA vaccines may improve protection by targeting additional antigens such as neuraminidase (NA), for which immune responses independently correlate with protection. In this phase 1/2 trial (NCT05333289), healthy adults 18–75 years were randomly assigned to receive different doses of mRNA-1020 or mRNA-1030 (encoding HA and NA at different ratios), mRNA-1010 (encoding HA), or a licensed active comparator (recombinant HA). Primary endpoints were safety and reactogenicity, and HA and NA antibody responses against vaccine-matched influenza strains. Most common local and systemic solicited ARs were injection site pain and fatigue. There were no vaccine-related serious adverse events nor significant associated safety concerns through 181 days. mRNA-1020 and mRNA-1030 elicited high HA-specific immune responses and induced NA-specific immune responses with no additional reactogenicity at equivalent dose levels beyond an mRNA-based, HA-only–containing vaccine.

Despite the availability of vaccines, seasonal influenza remains a substantial public health burden worldwide[1], with an estimated 3.2 million influenza-associated hospitalizations[2] and 291,000 to 646,000 respiratory deaths annually[3]. Currently available seasonal influenza vaccines utilize egg-, cell-, or recombinant protein-based platforms[4] that primarily target the hemagglutinin (HA) glycoprotein[5] and have a suboptimal effectiveness of 40% to 60%, even in seasons when most circulating influenza viruses are well matched to those included in the vaccines[6]. Improved influenza vaccine designs are thus needed to increase effectiveness and reduce disease burden.

Improving the neuraminidase (NA) content of seasonal influenza vaccines has been identified as a major area to increase effectiveness[5,7,8].

HA and NA are the two most abundant surface glycoproteins of influenza A and B viruses[9] that cause seasonal influenza epidemics in humans[1,10], and these proteins play a vital role in viral entry and release[11]. Currently, recombinant protein-based influenza vaccines only contain HA, while egg- and cell-based influenza vaccines contain both HA and NA[4], but only HA content is standardized[5]. Therefore, the protection provided by current vaccines is primarily mediated by antibodies against HA[5], with the NA content of current vaccines being absent or suboptimal, resulting in a lack of NA-based immunity among vaccine recipients[5,7,12–15]. However, individuals infected with the influenza virus generate inhibitory antibodies against both HA and NA[5,7], and antibody responses to NA provide protection against influenza in an independent manner from HA-based immunity[12,16–18]. Further, anti-NA antibodies are

[1]Moderna, Inc., 325 Binney Street, Cambridge, MA, USA. [2]Velocity Clinical Research, 2230 Auburn Avenue, Cincinnati, OH, USA. [3]CCT Research, 5740 Crestwood Dr, South Ogden, UT, USA. [4]Meridian Clinical Research, 4802 Sunnybrook Drive, Sioux City, IA, USA. ✉e-mail: Raffael.Nachbagauer@modernatx.com

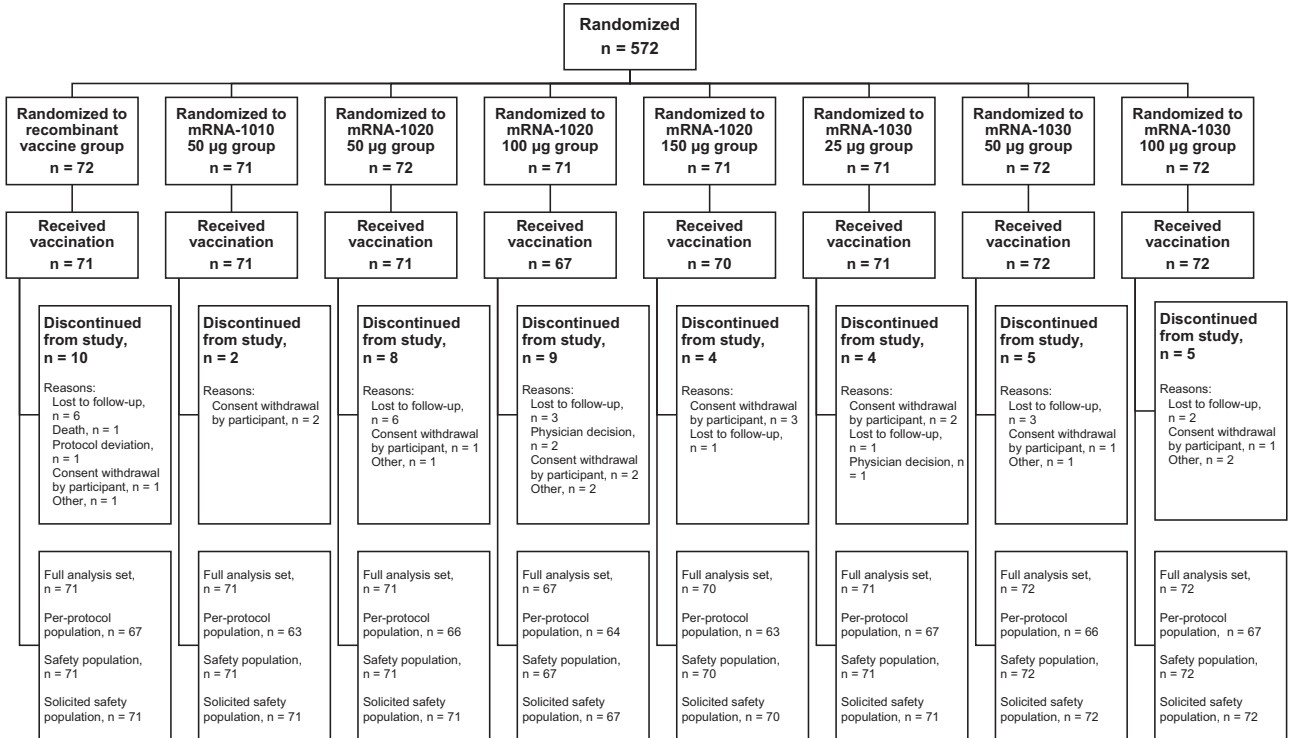

**Fig. 1 | Participant disposition.** The full analysis set consisted of all randomly assigned participants who received the investigational product. The per-protocol population consisted of all participants in the full analysis set who complied with the injection schedule and the timings of immunogenicity blood sampling to have a baseline and at least one post-injection assessment, did not have influenza infection at baseline through day 29, and had no major protocol deviations that impacted the immune response. The safety population consisted of all randomly assigned participants who received the investigational product. The solicited safety population consisted of all participants in the safety population who contributed any solicited adverse reaction data. Data cutoff date of May 24, 2023.

correlated with shortened viral shedding duration, as well as reduced duration, number, and severity of symptoms in individuals infected with influenza[17,18]. Since antibodies against HA and NA independently contribute to influenza protection, targeting both HA and NA may provide better protection by limiting the ability of the virus to escape immune responses through mutations and antigenic drift, which has been shown to occur in a discordant manner for HA and NA[19].

The mRNA platform has the potential to address limitations of current seasonal influenza vaccines due to its ability to include mRNAs encoding both HA and NA antigens. We previously demonstrated positive safety and immunogenicity findings for an mRNA-based seasonal influenza vaccine (mRNA-1010), which targets HA only[20–22]. Here, we report first-in-human findings from a phase 1/2 study of mRNA-based seasonal influenza vaccines targeting both HA and NA (mRNA-1020 and mRNA-1030) in healthy adults. Both vaccines contain mRNAs encoding for HA and NA of recommended seasonal influenza strains, but at different HA:NA ratios; the endogenous ratio of HA:NA on the surface of influenza viruses is estimated to be 6:1[23].

## Results
### Trial population
Participants were recruited into this study from March 31 to May 10, 2022. Safety follow-up continued through day 181. A total of 572 participants were randomly assigned to a vaccine group, of whom 565 participants received a vaccination and were included in the full analysis set. A total of 208, 215, 71, and 71 participants received mRNA-1020, mRNA-1030, mRNA-1010, and the recombinant vaccine, respectively (Fig. 1). Across all vaccine groups, 80.5% (455/565) of participants were White, 55.0% (311/565) of participants were female, and the median age of participants was 49 years (Supplementary Table 1).

### Safety
Any solicited AR, reflective of reactogenicity, within 7 days of vaccination was reported by 93.3% of mRNA-1020 recipients (any dose), 90.2% of mRNA-1030 recipients (any dose), 91.5% of mRNA-1010 (50 μg) recipients, and 74.6% of recombinant vaccine comparator recipients (Fig. 2). For all vaccine groups, the most common solicited local AR was injection site pain, and the most common solicited systemic AR was fatigue. For any solicited AR, the median day of onset was day 1, and the median duration ranged from 2 to 4 days. No participants reported a grade 4 solicited AR. Any grade 3 solicited ARs were reported by 12.7% (50 μg), 29.9% (100 μg), and 32.9% (150 μg) of mRNA-1020 recipients; 7.0% (25 μg), 20.8% (50 μg), and 27.8% (100 μg) of mRNA-1030 recipients; 12.7% of mRNA-1010 recipients (50 μg); and 0% of recombinant vaccine recipients.

Within 28 days of vaccination, unsolicited adverse events (AEs) were reported by 24.5% of mRNA-1020 recipients (any dose), 13.5% of mRNA-1030 recipients (any dose), 23.9% of mRNA-1010 recipients, and 14.1% of recombinant vaccine recipients (Supplementary Table 2). Few participants had AEs that were assessed as related to the vaccine by the investigator: mRNA-1020, 7.2%; mRNA-1030, 5.1%; mRNA-1010, 11.3%; recombinant vaccine, 1.4%; these AEs were primarily associated with reactogenicity, all were non-serious, and none were considered severe.

The median duration of safety follow-up from the day of vaccine administration ranged from 175 (mRNA-1030 group [25 μg]) to 179 days (mRNA-1030 group [100 μg]). One participant in the recombinant vaccine group died during the study from an accidental multi-drug overdose 105 days after vaccination, which was considered unrelated to study vaccination by the investigator. No other study discontinuations occurred due to an AE (Supplementary Table 2). Ten participants reported 12 severe AEs throughout the study, and none were considered related to study vaccination by the investigator. Serious AEs occurred in 12 participants during the

## A    Local Reactions

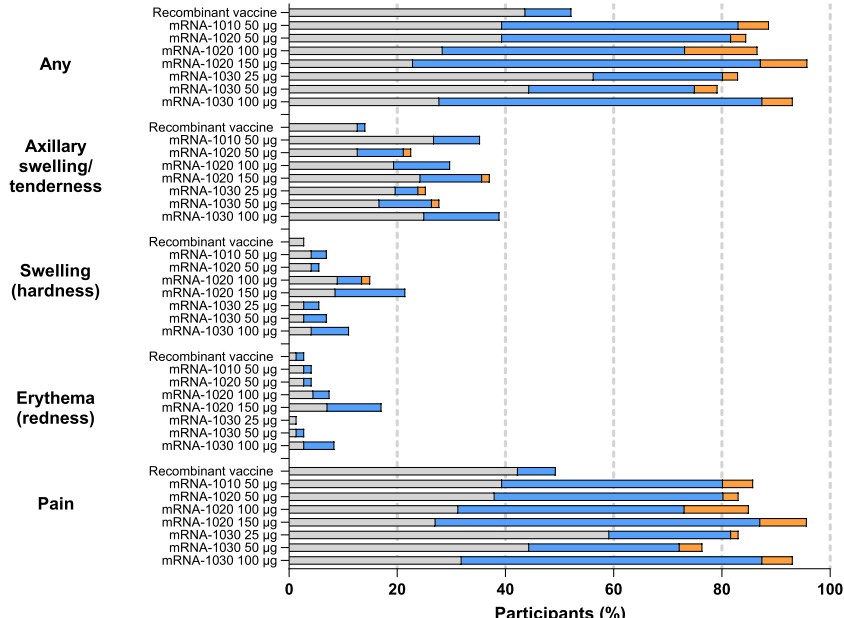

## B    Systemic Events

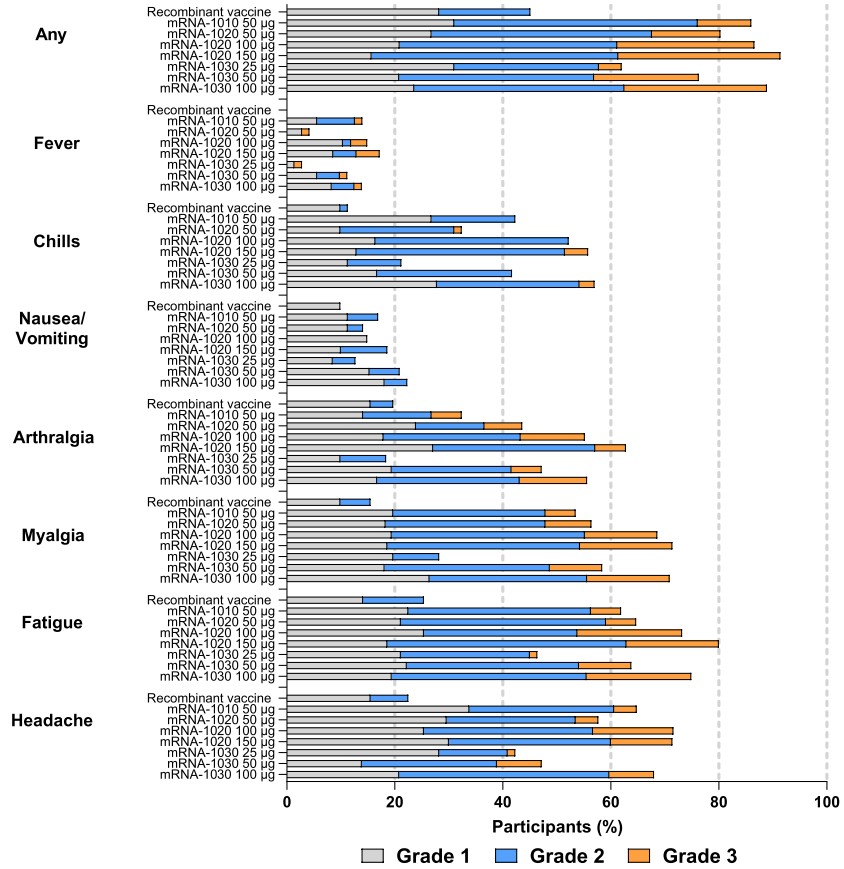

**Fig. 2 | Local (panel A) and systemic (panel B) solicited adverse reactions within 7 days after vaccination (solicited safety population).** Percentages of participants in the solicited safety population, which consisted of all participants in the safety population (all randomly assigned participants who received the investigational product) who contributed any solicited adverse reaction data: recombinant vaccine, $n = 71$; mRNA-1010 50 µg, $n = 71$, mRNA-1020 50 µg, $n = 71$; mRNA-1020 100 µg, $n = 67$; mRNA-1020 150 µg, $n = 70$; mRNA-1030 25 µg, $n = 71$; mRNA-1030 50 µg, $n = 72$; mRNA-1030 100 µg, $n = 72$. Adverse reactions were graded in intensity by the extent to which they affected the participant's daily activities as mild (grade 1), moderate (grade 2), severe (grade 3), or life-threatening (grade 4). When there were multiple adverse reactions of a specific term within 7 days, the worst grade was selected. In this study, no participants had a grade 4 solicited adverse reaction.

study; none were considered related to the vaccine by the investigator. Six participants had AESIs, of which 2 were assessed as related to the vaccine by the investigator: Bell's palsy (onset day 4; resolved day 43) in the mRNA-1020 100-μg group and thrombocytopenia (onset day 98; resolved day 104) in the mRNA-1030 25-μg group. All 4 participants with thrombocytopenia AESIs (3 assessed as unrelated) were asymptomatic, 2 had platelet counts below the AESI threshold of $<150 \times 10^9$/L prior to vaccination, and all events were detected during routine laboratory testing. Overall, 160 participants had MAAEs; most common were COVID-19 ($n = 33$), followed by upper respiratory tract infection ($n = 16$), and rhinovirus infection ($n = 10$). No participant was positive for influenza post-baseline, as tested by reverse transcription-polymerase chain reaction (RT-PCR). Four participants had MAAEs that were considered vaccine related by the investigator (Supplementary Table 2): blood pressure increased (mRNA-1020 50 μg; onset day 1; resolved day 2), Bell's palsy (mRNA-1020 100 μg; described previously), thrombocytopenia ($<150 \times 10^9$ platelet count; mRNA-1030 25 μg; described previously), and petechiae (normal platelet count; mRNA-1030 25 μg; onset day 3; resolved day 50).

## Immunogenicity

HAI antibody titers at baseline were similar across vaccine groups. A single dose of mRNA-1020 or mRNA-1030 at all evaluated dose levels elicited high HAI antibody titers against vaccine-matched influenza A (H1N1 and H3N2 subtypes) and B (Victoria- and Yamagata-lineage) strains at day 29 (Fig. 3) that remained above baseline at day 181 (Fig. 4). Seroconversion rates at day 29 were higher for A/H1N1 and A/H3N2 in comparison with B/Victoria and B/Yamagata within each vaccine group (Supplementary Table 3). The GMFRs from baseline at day 29 exceeded the four-fold threshold for both influenza A strains for all dosing groups. mRNA-1020 and mRNA-1030 generated HAI titers that were in a similar range to those elicited by the mRNA-1010 and recombinant vaccines.

NAI antibody titers were low for all vaccine groups at baseline. A single dose of mRNA-1020 or mRNA-1030 at all evaluated dose levels elicited high NAI antibody titers against vaccine-matched influenza A (H1N1 and H3N2 subtypes) and B (Victoria- and Yamagata-lineage) strains at day 29 (Fig. 5) that remained above baseline at day 181 (Fig. 4). NAI titers remained close to baseline in the HA-only recombinant vaccine and mRNA-1010 groups. The percentages of participants with a ≥ two-fold rise in NAI titers at day 29 were higher for B/Victoria and B/Yamagata in comparison with A/H1N1 and A/H3N2 within each of the mRNA-1020 and mRNA-1030 vaccine groups (Supplementary Table 4). Overall, mRNA-1020 and mRNA-1030 elicited HA and NA immune responses early at day 8 that peaked or were sustained at day 29 and remained above baseline at day 181 (Fig. 4, Supplementary Tables 3 and 4).

## Discussion

In this phase 1/2 clinical trial evaluating the safety, reactogenicity, and immunogenicity of two mRNA-based seasonal influenza vaccines (mRNA-1020 and mRNA-1030) targeting both HA and NA, we demonstrate the proof of concept that mRNA-1020 and mRNA-1030 elicit similar HA-specific immune responses to HA-only vaccines, while also inducing NA-specific immune responses with no additional reactogenicity beyond an mRNA-based, HA-only-containing vaccine. Through 6 months, no deaths or serious adverse events were related to vaccination with mRNA-1020 or mRNA-1030 at any dose. These findings support further development of mRNA vaccine approaches that simultaneously target both HA and NA surface glycoproteins of seasonal influenza viruses to potentially broaden protection against disease.

This was a first-in-human study of candidate mRNA vaccines simultaneously encoding NA in addition to HA antigens. Improving

NA content of seasonal influenza vaccines has been identified as a major focus to increase effectiveness[5,7,8], since current vaccines do not specifically elicit NA-based immunity[5,7,12–15] that could provide protection against influenza independent of HA-based immunity[12,16–18]. Here, we leveraged mRNA technology, which enables the encoding of multiple antigens in a single vaccine, to evaluate two candidate vaccines with different HA:NA ratios. Overall, a single dose of either mRNA vaccine candidate induced HAI titers that were similar to HA-only mRNA- or recombinant-based vaccine comparators for influenza A and B strains. The latter is a high-dose vaccine that is preferentially recommended for protection of influenza in adults aged ≥65 years[24]. Further, mRNA-1020 and mRNA-1030 elicited high NAI antibody titers at day 29, regardless of dose level, for which ≥two-fold rises have been shown to correspond to at least 30% protection against influenza illness[12,25]. HAI and NAI antibody titers elicited by mRNA-1020 and mRNA-1030 remained above baseline at day 181. mRNA-1020 and mRNA-1030 with the same HA content had similar GMFRs, suggesting no immune interference of NA towards HA. The lower observed fold-rises for N1 titers compared to other influenza strains may be partly due to a higher baseline in the assay, which reduced the sensitivity for detecting low increases from baseline. Comparatively lower titers against N1 have been previously observed[26]. Based on these findings, we are encouraged that mRNA-1020 and mRNA-1030 may provide additional protection. Targeting both HA and NA may also reduce the ability of influenza viruses to evade immunity through mutations and antigenic drift, which occur in an independent manner for HA and NA[19].

All dose levels of mRNA-1020 and mRNA-1030 were well tolerated, with solicited ARs most frequently reported at a maximum severity of grade 1 or grade 2. There were no grade 4 solicited ARs, and the incidence of grade 3 solicited ARs was lower for lower dose levels of mRNA-1020 and mRNA-1030. Overall, solicited ARs were more frequently reported in the mRNA vaccine groups than in the recombinant vaccine comparator group. mRNA-1020 and mRNA-1030 were found to have an acceptable safety profile. Two AESIs were considered related to study vaccination by the Investigator (Bell's palsy in the mRNA-1020 100-μg group; thrombocytopenia in the mRNA-1030 25-μg group); however, both cases had confounding factors and were assessed as not related to study vaccination by the study sponsor. Additional evaluation of mRNA-1020 and mRNA-1030 in larger clinical trials is warranted to further develop and assess the safety profile.

Study strengths include the randomized active comparator-controlled design that allowed for descriptive analysis to a licensed seasonal influenza vaccine. The small population size of US adults limited statistical comparisons across the vaccine groups and generalizability to other geographic regions.

These data provide the first demonstration of mRNA-based vaccines targeting both the HA and NA surface glycoproteins of four subtypes/lineages of seasonal influenza viruses in humans (A/H1N1, A/H3N2, B/Victoria, and B/Yamagata). Of note, inclusion of the B/Yamagata lineage in upcoming vaccine formulations is no longer recommended by WHO due to its global disappearance during the COVID-19 pandemic. These findings thus support the continued development of mRNA vaccines to combat clinically relevant seasonal influenza viruses (A/H1N1, A/H3N2, and B/Victoria), which remain a substantial public health burden worldwide.

## Methods
### Trial design and participants
This phase 1/2, randomized, observer-blind trial (ClinicalTrials.gov, NCT05333289) was conducted at 15 sites in the United States during the northern hemisphere spring of 2022 and was designed to evaluate the safety, reactogenicity, and immunogenicity of mRNA-1020 and mRNA-1030 candidate seasonal influenza vaccines in healthy adults.

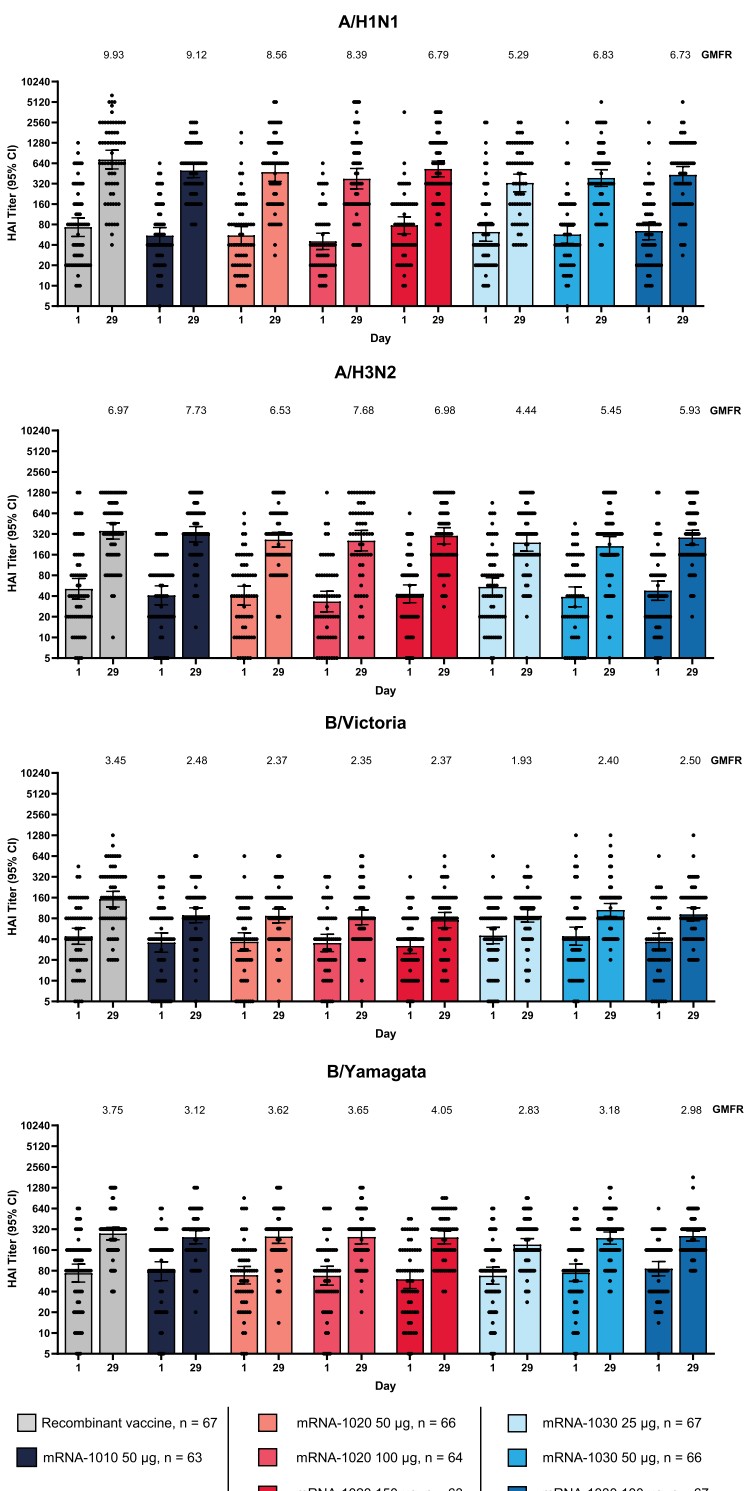

**Fig. 3 | Anti-hemagglutinin antibody geometric mean titer and geometric mean fold rise (per-protocol population).** HAI assay geometric mean titers against seasonal influenza A and B strains (A/Wisconsin/588/2019(H1N1)pdm09; A/Cambodia/e0826360/2020(H3N2); B/Washington/02/2019 [B/Victoria lineage]; B/Phuket/3073/2013 [B/Yamagata lineage]) were measured at day 1 (baseline) and day 29 (28 days after vaccination). Antibody titers <LLOQ were replaced by 0.5 × LLOQ. Titers >ULOQ were converted to the ULOQ. LLOQ were 10 for A/H1N1, A/H3N2, B/Victoria, and B/Yamagata. ULOQ were 1280 for A/H3N2, 3200 for B/Victoria, and 6400 for A/H1N1 and B/Yamagata. Dots correspond to participant-level titers. Error bars represent 95% confidence intervals. GMFR at day 29 from day 1 are shown above each day 29 bar plot. Number of participants (*n*) based on the per-protocol population, which consists of all participants who received study vaccine, complied with the timing of immunogenicity blood sampling to have a baseline and ≥1 post-baseline assessment, did not have influenza infection at baseline through day 29 (as documented by RT-PCR testing), and had no major protocol deviations that impacted the immune response. GMFR = geometric mean fold rise, HAI = hemagglutination inhibition, LLOQ = lower limit(s) of quantitation, RT-PCR = reverse transcription-polymerase chain reaction, ULOQ = upper limit(s) of quantitation.

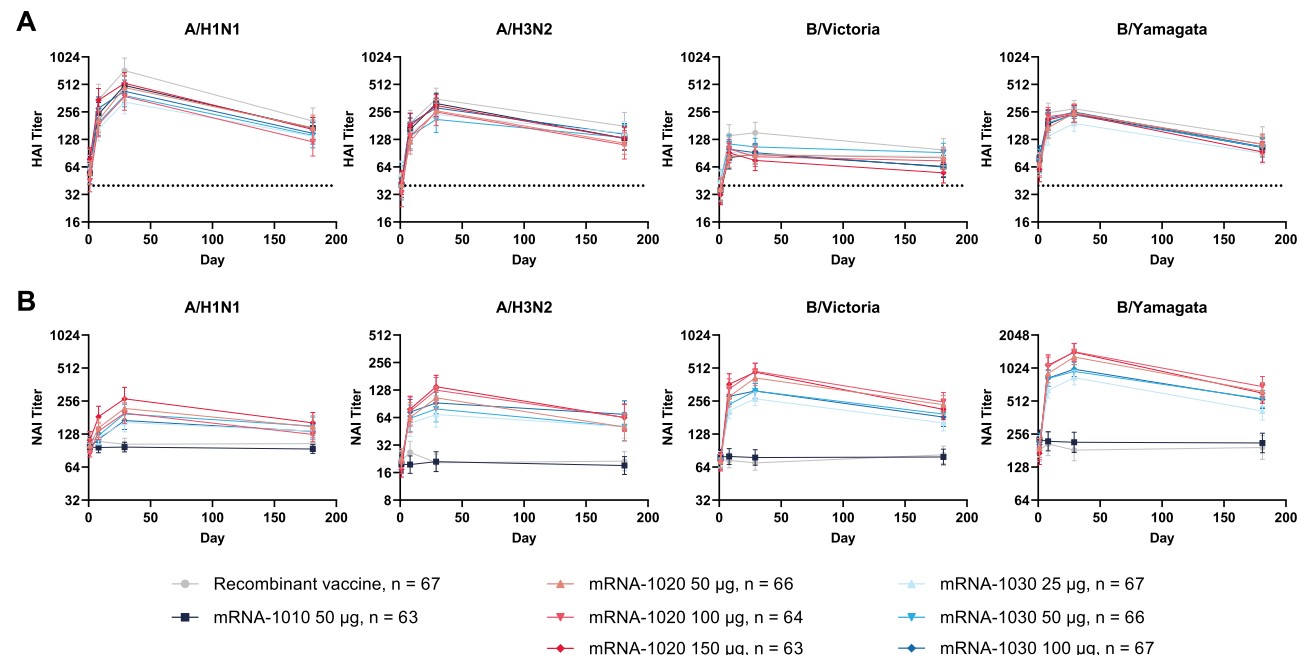

**Fig. 4 | Persistence of hemagglutination inhibition (panel A) and neuraminidase inhibition (panel B) responses through to day 181 (per-protocol population).** GMTs with associated 95% CI against vaccine-matched seasonal influenza A and B strains via the (A) HAI assay and (B) NAI assay at day 1 (baseline), day 8, day 29, and day 181. Horizontal dotted line indicates 1:40 titer associated with a 50% reduction in risk of infection. Number of participants (*n*) based on the per-protocol population, which consists of all participants who received study vaccine, complied with the timing of immunogenicity blood sampling to have a baseline and ≥1 post-baseline assessment, did not have influenza infection at baseline through day 29 (as documented by RT-PCR testing), and had no major protocol deviations that impacted the immune response. CI = confidence interval, GMT = geometric mean titer, HAI = hemagglutination inhibition, NAI = neuraminidase inhibition, RT-PCR = reverse transcription-polymerase chain reaction.

Medically stable adults aged 18–75 years with a body mass index of 18–35 kg/m² were eligible to participate in this study. Full inclusion and exclusion criteria are provided in the study protocol, available online with the full text of this article.

Eligible participants were randomly allocated in a 1:1:1:1:1:1:1:1 ratio to eight vaccine groups: mRNA-1020 at three dose levels (50, 100, or 150 μg), mRNA-1030 at three dose levels (25, 50, or 100 μg), an mRNA-based HA comparator (mRNA-1010 50 μg), or a licensed quadrivalent recombinant vaccine (Flublok, Sanofi Pasteur Inc., Bridgewater, NJ, USA) as an HA-only active comparator (Supplementary Fig. 1). Random allocation was stratified by age (18–49 vs. 50–75 years) to ensure balance of the two age groups within each vaccine group. The sponsor's biostatistics department or designee generated the randomized allocation schedule for vaccine group assignment using interactive response technology. Vaccine dose preparation and administration were performed by unblinded personnel who had no other role in the conduct of the trial.

This trial was conducted in accordance with the Declaration of Helsinki and Good Clinical Practice guidelines of the International Council for Harmonization, and the protocol was approved by the Advarra central institutional review board (protocol number Pro00061700). All participants provided written informed consent. Moderna, Inc., was responsible for the design of the trial and for the analysis of data.

## Vaccines
The mRNA vaccines tested in this study were lipid nanoparticle dispersions containing mRNA encoding surface glycoproteins of each of the four strains recommended by the World Health Organization for 2021/2022 northern hemisphere vaccines (A/Wisconsin/588/2019 [H1N1]; A/Cambodia/e0826360/2020 [H3N2]; B/Washington/02/2019 [B/Victoria lineage]; B/Phuket/3073/2013 [B/Yamagata lineage]). mRNA-1020 and mRNA-1030 each contained mRNA encoding four HA

and four NA antigens (HA:NA mass ratio of 1:1 and 3:1, respectively), whereas mRNA-1010 contained mRNA encoding four HA antigens only (Supplementary Table 5). The licensed recombinant vaccine (180 μg) contained four recombinant HA proteins (each 45 μg). All vaccines were administered at a 0.5 mL volume as a single intramuscular injection into the deltoid muscle on day 1.

## Trial objectives
The primary objectives of this study were to evaluate the safety and reactogenicity of mRNA-1020, mRNA-1030, and mRNA-1010 and to evaluate the antibody responses elicited by mRNA-1020, mRNA-1030, and mRNA-1010 against vaccine-matched influenza A and B strains at day 29. The secondary objective of this study was to evaluate the antibody responses elicited by mRNA-1020, mRNA-1030, and mRNA-1010 against vaccine-matched influenza A and B strains at all evaluable immunogenicity time points up to the end of the study (day 181).

## Safety
The primary safety/reactogenicity end points of this study were the frequency and grade of each solicited local and systemic adverse reaction (AR) within 7 days after vaccination; the frequency and severity of any unsolicited adverse events (AEs) within 28 days after vaccination; and the frequency of any AEs that were serious (SAEs), of special interest (AESI), medically attended (MAAEs), or led to discontinuation from study participation from day 1 through to the end of the study (day 181). An eDiary solicited daily participant reporting of ARs using a structured checklist on the day of vaccine administration and on the following 6 days. Unsolicited AEs were recorded after vaccination at clinic visits and during safety phone calls throughout the study. Blood collection for safety laboratory testing was performed at screening (up to 28 days prior to vaccination) and on day 8. Participants with influenza-like illness symptoms were instructed to notify

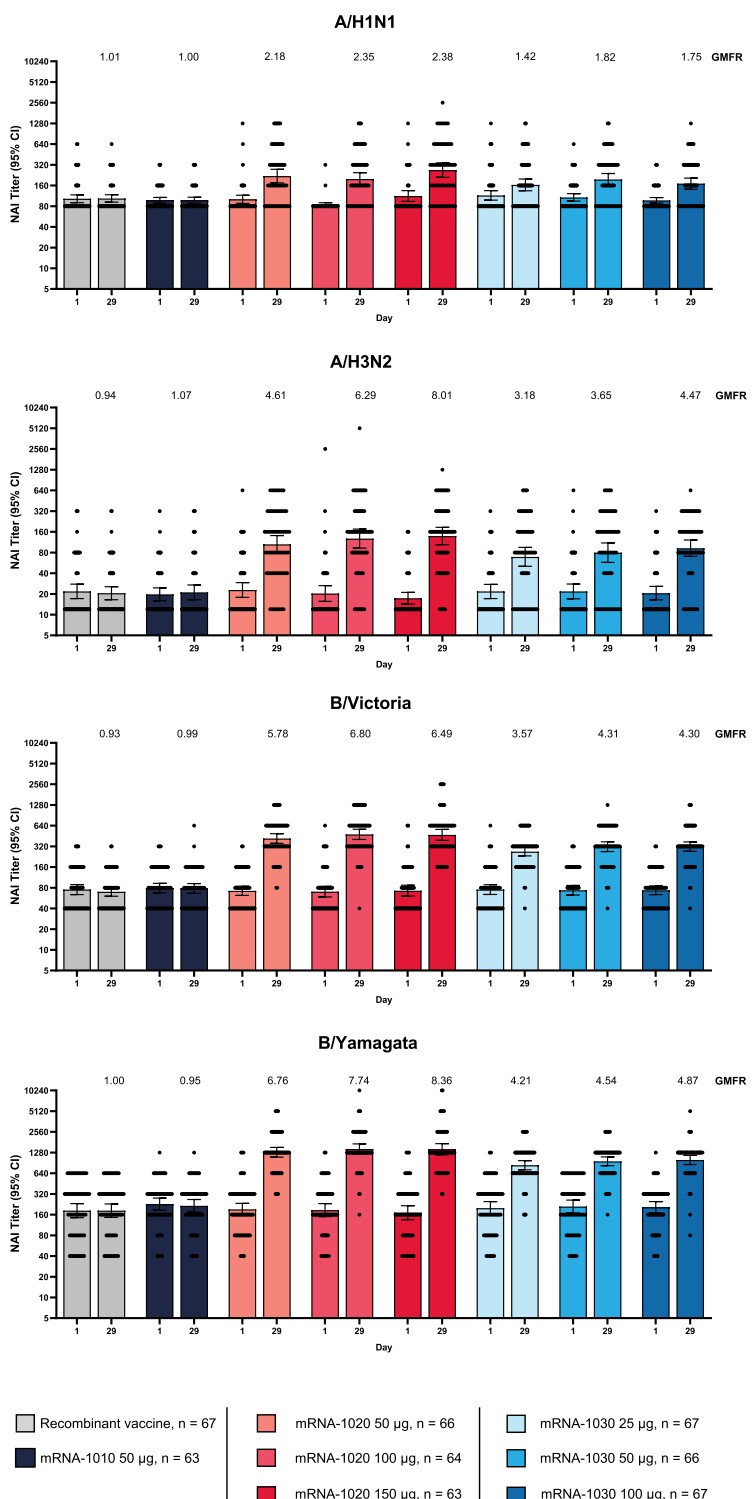

**Fig. 5 | Anti-neuraminidase antibody geometric mean titer and geometric mean fold rise (per-protocol population).** NAI assay geometric mean titers against seasonal influenza A and B strains (A/Wisconsin/588/2019 (H1N1)pdm09; A/Cambodia/e0826360/2020 (H3N2); B/Washington/02/2019 [B/Victoria lineage]; B/Phuket/3073/2013 [B/Yamagata lineage]) were measured at day 1 (baseline) and day 29 (28 days after vaccination). Antibody titers <LLOQ were replaced by 0.5 × LLOQ. Titers >ULOQ were converted to the ULOQ. LLOQ were 24 for A/H3N2, 80 for B/Victoria and B/Yamagata, and 160 for A/H1N1. ULOQ were 12,177 for B/Victoria, 20,480 for A/H3N2, 40,960 for A/H1N1, and 81,920 for B/Yamagata. Dots correspond to participant-level titers. Error bars represent 95% confidence intervals. GMFR at day 29 from day 1 are shown above each day 29 bar plot. Number of participants (*n*) based on the per-protocol population, which consists of all participants who received study vaccine, complied with the timing of immunogenicity blood sampling to have a baseline and ≥1 post-baseline assessment, did not have influenza infection at baseline through day 29 (as documented by RT-PCR testing), and had no major protocol deviations that impacted the immune response. GMFR = geometric mean fold rise, LLOQ = lower limit(s) of quantitation, NAI = neuraminidase inhibition, RT-PCR = reverse transcription-polymerase chain reaction, ULOQ = upper limit(s) of quantitation.

the site in order to undergo medical evaluation and nasal swabs for RT-PCR testing.

## Immunogenicity

Blood collection for immunogenicity was performed on days 1, 8, 29, and 181 (end of the study). Serum anti-HA antibody levels were measured by a qualified hemagglutination inhibition (HAI) assay, and NA-specific antibody levels were measured by a qualified NA inhibition (NAI) assay (Supplementary Methods). The primary immunogenicity end points included the geometric mean titer (GMT) of anti-HA and anti-NA antibodies against vaccine-matched influenza A and B strains at baseline (day 1) and day 29; geometric mean fold rise (GMFR) of anti-HA and anti-NA antibodies at day 29 versus baseline; the percentage of participants with seroconversion at day 29 (defined as a titer of ≥1:40 [if baseline was <1:10] or a ≥ four-fold rise in titer [if baseline was ≥1:10]) measured by the HAI assay; and the percentage of participants with ≥two-fold, ≥three-fold, and ≥four-fold rise in titers at day 29 as measured by the NAI assay. The secondary immunogenicity end points included the GMT and GMFR (versus baseline) of anti-HA or anti-NA antibodies at all evaluable time points.

## Statistical analyses

The sample size for this study was not driven by statistical assumptions for formal hypothesis testing. A total of approximately 560 participants, with 70 participants randomly assigned into each vaccine group, was planned and considered to be sufficient to provide descriptive safety and immunogenicity of different dose levels of mRNA-1020 or mRNA-1030. Sex of participants was not considered in the study design and was determined based on self-report.

The 95% CIs for GMT and GMFR were calculated based on the $t$ distribution of the log-transformed values, then back-transformed to the original scale. HAI seroconversion rate and the percentage of participants with ≥two-fold, ≥three-fold, and ≥four-fold rise in NAI titers were provided with a two-sided 95% CI using the Clopper-Pearson method. Antibody values below the lower limit of quantitation (LLOQ) were replaced by 0.5 × LLOQ. Antibody values above the upper limit of quantitation (ULOQ) were converted to the ULOQ. Missing safety and immunogenicity data were not imputed. Statistical analyses were performed using SAS version 9.4 or higher.

## Reporting summary

Further information on research design is available in the Nature Portfolio Reporting Summary linked to this article.

# Data availability

Access to participant-level data presented in this article and supporting clinical documents with external researchers who provide methodologically sound scientific proposals will be available upon reasonable request for products or indications that have been approved by regulators in the relevant markets and subject to review from 24 months after study completion. Such requests can be made to Moderna, Inc., 325 Binney Street, Cambridge, MA 02142 «data_sharing@modernatx.com». A materials transfer and/or data access agreement with the sponsor will be required for accessing shared data. All other relevant data are presented in the paper. The protocol is available online at ClinicalTrials.gov: NCT05333289.

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

## Acknowledgments

This work was supported by Moderna, Inc. Medical writing and editorial assistance (including the first draft) were provided by Renee Gordon, Ph.D., of MEDiSTRAVA in accordance with Good Publication Practice guidelines, under the direction of the authors, and funded by Moderna, Inc.

## Author contributions

I.T.L., K.K., K.S., J.H., D.E., D.S., A.P., R.P., J.A., and R.N. contributed to the study concept and design. Data was collected by I.T.L., K.K., K.S., A.A., D.K.L., J.H., D.E., D.S., and J.V. and analyzed and interpreted by A.K.R.S., I.T.L., K.K., K.S., A.A., J.H., D.E., D.S., B.H., A.P., J.V., R.P., J.A., and R.N. All authors contributed to the drafting and critical review of this manuscript and approved the final draft. The trial sponsor, Moderna, Inc., was involved in the conceptualization and trial design; collection, analysis, and interpretation of the data; preparation or approval of the manuscript; and the decision to submit the manuscript for publication.

## Competing interests

A.K.R.S., K.K., K.S., A.A., D.S., B.H., A.P., J.V., R.P., and R.N. are employees of Moderna, Inc., and may hold stock/stock options in the company. I.T.L. and J.A. were employees of Moderna, Inc., at the time of the study. D.E., D.K.L., and J.H. declare no competing interests.
