## [Peer Review file · Nature Communications]

Immunogenicity and safety of mRNA-based seasonal influenza vaccines encoding hemagglutinin and neuraminidase

Corresponding Author: Dr Raffael Nachbagauer

Version 0:

Reviewer comments:

Reviewer #1

(Remarks to the Author)

This manuscript is a well-written description of a Phase 1/2 study of a quadrivalent seasonal influenza vaccine based on mRNA technology. The design and analytic approach are appropriate and the sample size is adequate for the assessment of the primary endpoint of safety. The sample sizes of each treatment group and the formulation of the vaccines administered to each group are not, however, adequate to make a solid dose selection for further clinical development. The following specific comments are made:

- 1) The incidence of severe (grade 3) solicited adverse reactions following dose administration is quite high in comparison to conventional vaccine platforms - this should be commented upon and the possible impact on uptake by public should be addressed.
- 2) The formulation of each study vaccine (including the commercial comparator) and including the µg content of each antigen, should be reported. Indication of the total µg of antigen, when this may refer to 4 or 8 antigens does not allow a reasonable comparison of potency (even if the sample size were adequate). The rationale for dose selection should be described.

Reviewer #3

(Remarks to the Author)

The authors describe a first-in-human evaluation of mRNA vaccines encoding for hemagglutinin (HA) and neuraminidase (NA) antigens from A/H1N1, A/H3N2, B/Victoria, and B/Yamagata influenza strains in persons 18-75 years of age. The authors report on both the safety and immunogenicity of these vaccines in comparison to licensed recombinant influenza vaccine and an investigational mRNA vaccine encoding for only hemagglutinin. The authors major assumption is that the addition of neuraminidase specific immunity to influenza vaccine would afford more protection from influenza than current standard influenza vaccines. The authors provide evidence to support this in the introduction to the manuscript. Both mRNA vaccines tested induced both HA and NA immune responses for the vaccine antigens evaluated. As expected, the mRNA vaccines were more reactogenic with some dose related increases in reactogenicity observed.

Specific comments

Results

Lines 76-78: Refer to comment for Supplementary Table 1

Line 88: Add dosage level of vaccine for mRNA-1010 (50µg)

Lines 91-96: Refer to comments for Supplementary Table 2. Consider adding a comment that none of the AEs within 28 days related to vaccination were severe.

Lines 110-112: Were any influenza specific illnesses reported?

Discussion

Lines 145-147: Comment on increased reactogenicity of the mRNA vaccines as compared to the recombinant vaccine. Also comment on the apparent dose dependent increase in reactogenicity. Ultimately the tolerability and acceptability of an

mRNA influenza vaccine will have an impact on annual vaccine uptake.

Lines 147-149: Please comment on the AESIs particularly the case of Bell's palsy that was observed. May not be fair to say there were no notable safety concerns at any dose. Would omit that phrase.

Lines 180-182: Would note that future development of seasonal influenza vaccines will not include B/Yamagata lineage which is currently not circulating.

Methods

Were participants followed for influenza illness?

Were any analyses conducted based on age of the participants e.g. stratified analysis of immune responses based on age < 65 years ≥ 65 years. Please report in methods and results if conducted.

Supplementary Table 1

Suggest adding total column for the study population as results noted in Lines 76-78 are described that way.

Supplementary Table 2

Suggest adding total columns for mRNA-1020 and mRNA-1030 as results described that way.

Reviewer #4

(Remarks to the Author)

This article reports on first-in-human findings of mRNA platform for seasonal influenza vaccines from a phase 1/2 study. The two vaccines target HA and NA. The paper reports on 565 participants randomized to receive mRNA-1020, mRNA-1030, mRNA-1010 and the recombinant vaccine. The primary endpoints were safety, reactogenicity and immunogenicity.

The data were appropriately displayed and analyzed.

The information in the paper is important.

Version 1:

Reviewer comments:

Reviewer #1

(Remarks to the Author)

The authors have responded to most of the initial reviewer comments quite adequately. The main issue (to which the authors declined to respond) is the actual doses of HA or NA in each of the exploratory vaccine formulations. Stating that this is "proprietary information" seems a bit coy, since the primary endpoint of safety appears to be strongly associated with dose. This may be an important factor in the ultimate assessment of the risk/benefit ratio. If this is acceptable to editors, the manuscript is otherwise a useful contribution.

Reviewer #3

(Remarks to the Author)

Reviewer comments addressed appropriately

REVIEWER COMMENTS

Reviewer #1

This manuscript is a well-written description of a Phase 1/2 study of a quadrivalent seasonal influenza vaccine based on mRNA technology. The design and analytic approach are appropriate and the sample size is adequate for the assessment of the primary endpoint of safety. The sample sizes of each treatment group and the formulation of the vaccines administered to each group are not, however, adequate to make a solid dose selection for further clinical development. The following specific comments are made:

- 1) The incidence of severe (grade 3) solicited adverse reactions following dose administration is quite high in comparison to conventional vaccine platforms - this should be commented upon and the possible impact on uptake by public should be addressed.

Response: The Discussion section was updated to further comment on the reactogenicity profile of the mRNA-1020 and mRNA-1030 vaccines. We do acknowledge that mRNA-1020 and mRNA-1030 vaccination in this phase 1/2 trial was associated with increased grade 3 solicited adverse reactions (ARs) relative to the recombinant vaccine comparator. However, the vaccines also elicited robust HAI and NAI titers, thus suggesting potential to broaden protection against influenza strains. Overall, additional evaluation of these vaccines is indeed warranted to better understand the benefit-risk profile.

Revised text (Discussion):

All dose levels of mRNA-1020 and mRNA-1030 were well tolerated, with solicited ARs most frequently reported at a maximum severity of grade 1 or grade 2. There were no grade 4 solicited ARs, and the incidence of grade 3 solicited ARs was lower for lower dose levels of mRNA-1020 and mRNA-1030. Overall, solicited ARs were more frequently reported in the mRNA vaccine groups than in the recombinant vaccine comparator group. mRNA-1020 and mRNA-1030 were found to have an acceptable safety profile. Two AESIs were considered related to study vaccination by the Investigator (Bell's palsy in the mRNA-1020 100 µg group; thrombocytopenia in the mRNA-1030 25 µg group); however, both cases had confounding factors and were assessed as not related to study vaccination by the study sponsor. Additional evaluation of mRNA-1020 and mRNA-1030 in larger clinical trials is warranted to further develop and assess the safety profile.

- 2) The formulation of each study vaccine (including the commercial comparator) and including the µg content of each antigen, should be reported. Indication of the total µg of antigen, when this may refer to 4 or 8 antigens does not allow a reasonable comparison of potency (even if the sample size were adequate). The rationale for dose selection should be described.

Response: Thank you for this comment. Please note that the vaccine components are detailed in the Methods section. mRNA-1020 and mRNA-1030 each contain mRNA encoding four HA and four NA antigens with a HA:NA mass ratio of 1:1 (mRNA-1020) and 3:1 (mRNA-1030). Within the HA and NA content respectively, the mRNA amount for each component was matched. Unfortunately, further details on the antigen compositions cannot be disclosed because it is proprietary information. However, full details are not required for the interpretation of phase 1/2 study results. For the commercial comparator Flublok, the manuscript was updated to specify the antigen content of the vaccine used in this study.

Revised text (Methods): *mRNA-1020 and mRNA-1030 each contained mRNA encoding four HA and four NA antigens (HA:NA mass ratio of 1:1 and 3:1, respectively), whereas mRNA-1010 contained mRNA encoding four HA antigens only. The licensed recombinant vaccine (180 µg) contained four recombinant HA proteins (each 45 µg).*

Reviewer #2

The authors describe a first-in-human evaluation of mRNA vaccines encoding for hemagglutinin (HA) and neuraminidase (NA) antigens from A/H1N1, A/H3N2, B/Victoria, and B/Yamagata influenza strains in persons 18-75 years of age. The authors report on both the safety and immunogenicity of these vaccines in comparison to licensed recombinant influenza vaccine and an investigational mRNA vaccine encoding for only hemagglutinin. The authors major assumption is that the addition of neuraminidase specific immunity to influenza vaccine would afford more protection from influenza than current standard influenza vaccines. The authors provide evidence to support this in the introduction to the manuscript. Both mRNA vaccines tested induced both HA and NA immune responses for the vaccine antigens evaluated. As expected, the mRNA vaccines were more reactogenic with some dose related increases in reactogenicity observed.

Specific comments

Results

1. Lines 76-78: Refer to comment for Supplementary Table 1

Response: Supplementary Table 1 was updated to have a column reporting demographics and characteristics of the overall study population (N = 565).

2. Line 88: Add dosage level of vaccine for mRNA-1010 (50µg)

Response: The text was updated accordingly to state 50 µg.

Revised text (Results): Any grade 3 solicited ARs were reported by 12.7% (50 µg), 29.9% (100 µg), and 32.9% (150 µg) of mRNA-1020 recipients; 7.0% (25 µg), 20.8% (50 µg), and 27.8% (100 µg) of mRNA-1030 recipients; 12.7% of mRNA-1010 50 µg recipients; and 0% of recombinant vaccine recipients.

3. Lines 91-96: Refer to comments for Supplementary Table 2. Consider adding a comment that none of the AEs within 28 days related to vaccination were severe.

Response: Thank you for this suggestion. The text was updated accordingly to indicate none of the AEs within 28 days related to vaccination were considered severe.

Revised text (Results): *Few participants had AEs that were assessed as related to the vaccine by the investigator: mRNA-1020, 7.2%; mRNA-1030, 5.1%; mRNA-1010, 11.3%; recombinant vaccine, 1.4%; these AEs were primarily associated with reactogenicity, all were non-serious, and none were considered severe.*

4. Lines 110-112: Were any influenza specific illnesses reported?

Response: No participant was RT-PCR positive for influenza post-Baseline. We have updated the manuscript to note this.

Revised text (Results): No participant was positive for influenza post-Baseline, as tested by reverse transcription-polymerase chain reaction (RT-PCR).

Discussion

5. Lines 145-147: Comment on increased reactogenicity of the mRNA vaccines as compared to the recombinant vaccine. Also comment on the apparent dose dependent increase in reactogenicity. Ultimately the tolerability and acceptability of an mRNA influenza vaccine will have an impact on annual vaccine uptake.

Response: As per our response to Reviewer 1, comment 1, the Discussion section was updated to further comment on the reactogenicity profile of the mRNA-1020 and mRNA-1030 vaccines. This was a proof-of-concept phase 1/2 study and therefore further evaluation of mRNA-1020 and mRNA-1030 versus licensed influenza vaccines in larger clinical trials is warranted before any comment on vaccine uptake can be made.

Revised text (Discussion): All dose levels of mRNA-1020 and mRNA-1030 were well tolerated, with solicited ARs most frequently reported at a maximum severity of grade 1 or grade 2. There were no grade 4 solicited ARs, and the incidence of grade 3 solicited ARs was lower for lower dose levels of mRNA-1020 and mRNA-1030. Overall, solicited ARs were more frequently reported in the mRNA vaccine groups than in the recombinant vaccine comparator group. mRNA-1020 and mRNA-1030 were found to have an acceptable safety profile. Two AESIs were considered related to study vaccination by the

Investigator (Bell's palsy in the mRNA-1020 100 µg group; thrombocytopenia in the mRNA-1030 25 µg group); however, both cases had confounding factors and were assessed as not related to study vaccination by the study sponsor. Additional evaluation of mRNA-1020 and mRNA-1030 in larger clinical trials is warranted to further develop and assess the safety profile.

6. Lines 147-149: Please comment on the AESIs particularly the case of Bell's palsy that was observed. May not be fair to say there were no notable safety concerns at any dose. Would omit that phrase.

Response: While we have accordingly removed the 'no notable safety concerns' phrase from the Discussion, we have also further clarified that the 2 AESIs (Bell's Palsy and thrombocytopenia) were considered by the Investigator to be related to vaccination, but these events were considered unrelated to vaccination by the Sponsor.

Revised text (Discussion): Two AESIs were considered related to study vaccination by the Investigator (Bell's palsy in the mRNA-1020 100 µg group; thrombocytopenia in the mRNA-1030 25 µg group); however, both cases had confounding factors and were assessed as not related to study vaccination by the study sponsor. Additional evaluation of mRNA-1020 and mRNA-1030 in larger clinical trials is warranted to further develop and assess the safety profile.

7. Lines 180-182: Would note that future development of seasonal influenza vaccines will not include B/Yamagata lineage which is currently not circulating.

Response: The text was updated accordingly.

Revised text (Discussion): These data provide the first demonstration of mRNA-based vaccines targeting both the HA and NA surface glycoproteins of four subtypes/lineages of seasonal influenza viruses in humans (A/H1N1, A/H3N2, B/Victoria, and B/Yamagata). Of note, inclusion of the B/Yamagata lineage in upcoming vaccine formulations is no longer recommended by WHO due to its global disappearance during the COVID-19 pandemic. These findings thus support the continued development of mRNA vaccines to combat clinically relevant seasonal influenza viruses (A/H1N1, A/H3N2, and B/Victoria), which remain a substantial public health burden worldwide.

Methods

8. Were participants followed for influenza illness?

Response: Participants with Influenza-like illness (ILI) symptoms were instructed to notify the site in order to undergo medical evaluation and nasal swabs, but no participants were RT-PCR positive for influenza post-Baseline. We have accordingly updated the Methods section and the Results section to address the Reviewer's comments.

Revised text (Methods): Participants with influenza-like illness symptoms were instructed to notify the site in order to undergo medical evaluation and nasal swabs for RT-PCR testing.

Revised text (Results): No participant was positive for influenza post-Baseline, as tested by reverse transcription-polymerase chain reaction (RT-PCR).

9. Were any analyses conducted based on age of the participants e.g. stratified analysis of immune responses based on age < 65 years ≥ 65 years. Please report in methods and results if conducted.

Response: We thank the reviewer for their interest. Participants in this study were not stratified by age <65 and ≥65 years and thus immune response analyses were not conducted in those age groups; therefore, the manuscript was not updated based on this comment.

Supplementary Table 1

10. Suggest adding total column for the study population as results noted in Lines 76-78 are described that way.

Response: Supplementary Table 1 was updated accordingly to have a column reporting demographics and characteristics of the overall study population (N = 565).

Supplementary Table 2

11. Suggest adding total columns for mRNA-1020 and mRNA-1030 as results described that way.

Response: Supplementary Table 2 was updated accordingly to have columns reporting adverse events among all mRNA-1020 recipients (any dose; n = 208) and among all mRNA-1030 recipients (any dose; n = 215).

Reviewer #3

This article reports on first-in-human findings of mRNA platform for seasonal influenza vaccines from a phase 1/2 study. The two vaccines target HA and NA. The paper reports on 565 participants randomized to receive mRNA-1020, mRNA-1030, mRNA-1010 and the recombinant vaccine. The primary endpoints were safety, reactogenicity and immunogenicity.

The data were appropriately displayed and analyzed.

The information in the paper is important.

Response: We thank the reviewer for their kind feedback.

EDITORIAL REQUESTS

Policies and Forms Required for Resubmission

* Please complete or update the following checklist(s) to verify compliance with our research ethics and data reporting standards. Address all points on the checklist, revising your manuscript in response to the points if needed.

The form(s) must be downloaded and completed in Adobe Reader rather than opened in a web browser. Each form must be uploaded as a Related Manuscript file at the time of resubmission.

Editorial policy checklist:

<https://www.nature.com/documents/nr-editorial-policy-checklist.pdf>

Reporting summary:

Response: We confirm that the editorial policy checklist and reporting summary are uploaded as Related Manuscript files.

* Nature journals have recently announced an update to our guidance on reporting on sex and gender in research studies (see here). We strongly encourage researchers to follow the 'Sex and Gender Equity in Research – SAGER – guidelines' and to include sex and gender considerations for studies involving humans, vertebrate animals and cell lines where relevant to the topic of study (an overview can be found here). Authors should use the terms sex (biological attribute) and gender (shaped by social and cultural circumstances) carefully in order to avoid confusing both terms.

When preparing your revised manuscript, please be aware of our guidance on Sex and Gender reporting).

Please note that we require that the following recommendations from the guidelines are followed:

1. If the research findings apply to only one sex or gender, that must be indicated in the title and/or abstract.

2a. For studies involving vertebrates animal and cell lines- The Reporting Summary should include whether sex was considered in the study design.

2b. For studies involving human research participants- The Reporting Summary should include whether sex and/or gender was considered in the study design and whether sex and/or gender of participants was determined based on self-report or assigned (and methodology used).

3. Data should be reported disaggregated for sex and gender where this information has been collected and consent has been obtained for reporting and sharing individual-level data; disaggregated numbers for individual experiments must be provided in the source data as appropriate whereas overall numbers may be provided in the Nature Portfolio Reporting Summary. Information on the points above should be included in the revised manuscript and detailed in the cover letter.

In addition, please note that if sex- and gender-based analyses have been performed a priori, results should be reported regardless of positive or negative outcome. We discourage conducting

post hoc sex- and gender-based analysis if the study design is insufficient (for example, low sample size) to enable meaningful conclusions.

If no sex- and gender-based analyses have been performed, please indicate the reasons for the lack of these analyses in the Reporting Summary.

Response: The sex of participants was not considered in the study design and was determined based on self-report. This detail has been added to the Methods section of the manuscript.

The number of participants by sex (male/female) who were randomly assigned to each vaccine group are reported in Supplementary Table 1.

Sex-based analyses were not planned in the study design/statistical analysis plan, and are therefore not reported in this manuscript.

Revised text (Methods): *Sex of participants was not considered in the study design and was determined based on self-report.*

Data and Code Availability

* All Nature Communications manuscripts must include a “Data Availability” section after the Methods section but before the References. If any of the data can only be shared on request or are subject to restrictions, please specify the reasons and explain how, when, and by whom the data can be accessed. For more information on this policy and a list of examples, see:

<https://www.nature.com/documents/nr-data-availability-statements-data-citations.pdf>

* As Nature Portfolio policies strongly encourage you to share your research data in a public repository (e.g. spreadsheets, text, images), we are partnering with the figshare repository so that you can use the figshare integration via the ‘Research Data Deposition’ tab when submitting your revised manuscript.

Data are stored privately until a manuscript decision is reached and you can edit/withdraw them up to this point: you retain rights and control over your data. The data will be published at the same time as your article; you will receive a data DOI, with guidance on linking the data and manuscript.

In the event your manuscript is not accepted, you can keep or remove your data in figshare.

We recommend the use of discipline-specific repositories where available and for a number of data types this is mandatory. Ensure you do not submit these data types or any sensitive data to figshare.

* We strongly encourage you to deposit all new data associated with the paper in a persistent repository where they can be freely and enduringly accessed. We recommend submitting the data to discipline-specific and community-recognised repositories; a list of repositories is provided here:

<http://www.nature.com/sdata/policies/repositories>

Refer to our data policies here: <https://www.nature.com/nature-portfolio/editorial-policies/reporting-standards#availability-of-data>

* To maximise the reproducibility of research data, we ask that you provide a Source Data file containing the raw data underlying the following types of display items:

- Any reported means/averages in box plots, bar charts, and tables

- Dot plots/scatter plots, especially when there are overlapping points
- Line graphs
- Uncropped and unprocessed scans of all blots and gels including all quantified replicates. The edge of membranes, molecular weight ladders and loading controls should be presented on all blots. Where membranes have been cut, please ensure that at least one marker above and below is present. For an example of presentation of full scan blots, see the Source Data file of <https://www.nature.com/articles/s41467-020-16984-1#Sec35> and for more information, please refer to <https://www.nature.com/nature-research/editorial-policies/image-integrity>.

The data should be provided in a single Excel file with data for each figure/table in a separate sheet, or in multiple labelled files within a zipped folder. Name this file or folder 'Source Data', and include a brief description in your cover letter. The "Data Availability" section should also include the statement "Source Data are provided with this paper."

To learn more about our motivation behind this policy, please see:

<https://www.nature.com/articles/s41467-018-06012-8>

A Source Data file is not necessary if all display items presented in the main manuscript and supplementary information can be reproduced from raw data and code that have already been shared in a public repository.

Response: A Data Availability statement is included in the manuscript, as well as here for your reference.

Access to participant-level data presented in this article and supporting clinical documents with external researchers who provide methodologically sound scientific proposals will be available upon reasonable request for products or indications that have been approved by regulators in the relevant markets and subject to review from 24 months after study completion. Such requests can be made to Moderna Inc., 325 Binney Street, Cambridge, MA 02142

[<<data_sharing@modernatx.com>>](mailto:data_sharing@modernatx.com). A materials transfer and/or data access agreement with the sponsor will be required for accessing shared data. All other relevant data are presented in the paper. The protocol is available online at ClinicalTrials.gov: NCT05333289.

* Please replace your bar graphs with plots that feature information about the distribution of the underlying data. All data points should be shown for plots with a sample size less than 10. For larger sample sizes, please consider box-and-whisker or violin plots as alternatives. Measures of centrality, dispersion and/or error bars should be plotted and described in the figure legend.

Response: Note that the sample size exceeds 10 in each study group.

For Figures 3 and 5, the bar graphs show measures of centrality (the geometric mean titer), as well as error bars showing 95% confidence intervals to indicate dispersion of data; this is

described in the respective figure legends. Therefore, no updates were made in response to this comment.

ORCID

* Nature Communications is committed to improving transparency in authorship. As part of our efforts in this direction, we are now requesting that all authors identified as 'corresponding author' create and link their Open Researcher and Contributor Identifier (ORCID) with their account on the Manuscript Tracking System prior to acceptance. ORCID helps the scientific community achieve unambiguous attribution of all scholarly contributions.

You can create and link your ORCID from the home page of the Manuscript Tracking System by clicking on 'Modify my Springer Nature account' and following these instructions.

Please also inform all co-authors that they can add their ORCIDs to their accounts and that they must do so prior to acceptance.

If you experience problems in linking your ORCID, please contact the Platform Support Helpdesk.

Response: We confirm that ORCID are provided for corresponding and co-authors.

Author Changes on Revision

If there are any changes to the author list in the revised manuscript, please use this approval form www.nature.com/documents/nr-author-list-change-form.pdf, arranging for all authors on your paper to sign the statement confirming that they agree to the author list being changed, and add this document to your resubmission.

Response: No changes have been made to the author list.

REVIEWER COMMENTS

Reviewer #1

- 1) The authors have responded to most of the initial reviewer comments quite adequately. The main issue (to which the authors declined to respond) is the actual doses of HA or NA in each of the exploratory vaccine formulation. Stating that this is "proprietary information" seems a bit coy, since the primary endpoint of safety appears to be strongly associated with dose. This may be an important factor in the ultimate assessment of the risk/benefit ratio. If this is acceptable to editors, the manuscript is otherwise a useful contribution.

Response: As requested, we have accordingly provided a supplementary table (Supplementary Table 5) that details the doses of HA and NA content for each of the mRNA study vaccines.